# Frequency-Agile FFT Spectrometer for Microwave Remote Sensing Applications

**Jonas Hagen *** [ID]**, Andres Luder, Axel Murk** [ID] **and Niklaus Kämpfer**

Institute of Applied Physics, University of Bern, 3012 Bern, Switzerland; andres.luder@iap.unibe.ch (A.L.);
axel.murk@iap.unibe.ch (A.M.); niklaus.kaempfer@iap.unibe.ch (N.K.)

**\*** Correspondence: jonas.hagen@iap.unibe.ch

**Abstract:** We report on a Fast Fourier Transform Spectrometer (FFTS) that provides larger bandwidth by fast local oscillator switching of the base-band converter. We demonstrate that this frequency scanning technique is suited for atmospheric remote sensing and conduct measurements of atmospheric ozone using the WIRA-C (WInd RAdiometer for Campaigns) Doppler wind radiometer. The comparison of our measurements to an adjusted atmospheric and instrumental model exposes no systematic biases due to the switching procedure in the measured spectra. It further shows that the combination of high spectral resolution with large bandwidth yields good measurement response to stratospheric and mesospheric ozone from approximately a 20 km to 70 km altitude with a resolution of 7 km in the lower stratosphere to 20 km in the mesosphere. We conclude that low-cost, low-power software-defined radio hardware designed for communications applications is very well suited for a variety of spectroscopic applications, including ozone monitoring. This allows the design of low-cost, multi-purpose instruments for atmospheric remote sensing and thus has a direct impact on future radiometer developments and their adoption in remote sensing campaigns and networks.

**Keywords:** radiometry; remote sensing; FFT spectrometry; ozone

## 1. Introduction

Spectrometers are used for the real-time observation of emission lines in remote sensing of the atmosphere and radio-astronomy. Fast Fourier Transform Spectrometers (FFTS) in particular are widely used in microwave remote sensing to observe the spectral distribution of thermal microwave emission of atmospheric trace gases. Compared to other spectrometer types, like optoacoustic spectrometers or filter-banks, FFTS offer a good compromise between bandwidth, spectral resolution, and stability [1,2]. An overview of different FFTS implementations and their performance characteristics was given by [3]. Lately, less expensive and more flexible radio peripherals became available as Software-Defined Radio (SDR), which not only allow the implementation of an FFTS, but also offer some flexibility in implementing further signal processing and acquisition schemes.

The WIRA-C (WInd RAdiometer for Campaigns) Doppler microwave wind radiometer [4] is a ground-based passive radiometer, which observes the ozone rotational emission line at 142.17504 GHz with a high spectral resolution of 12 kHz. It uses the Doppler shift of the emission line due to moving air to retrieve information about the horizontal wind speed in the stratosphere and lower mesosphere (approximately 30 to 70 km). The pressure broadening effect allows the retrieval of altitude-resolved wind profiles.

The spectrometer used in this instrument is a general purpose SDR that is available at a low cost and small form-factor and implements an FFTS. The trade-off is the relatively low bandwidth of 200 MHz, which is enough for wind observations in the stratosphere and lower mesosphere, but not for ozone observations in the lower stratosphere. Extending the bandwidth, while keeping the high

resolution at the line-center, would thus enable simultaneous ozone and wind measurements, among other novel applications.

Multi-purpose ground-based passive instruments that observe different species at the same time provide very valuable observations to understand the small scale variability and local behavior of dynamics and chemistry in the middle atmosphere. An instrument that observes multiple emission lines with a frequency switched double-side-band receiver has been shown to be valuable for simultaneous measurements of carbon monoxide and ozone [5]. In [6], the value of joint ozone and wind measurements was demonstrated at the arctic station in Ny-Ålesund, Svalbard, using the 110 GHz ozone emission line.

With this study, we present a method to extend the observation bandwidth of our spectrometer from 200 MHz to 1 GHz, which makes it possible to use our wind radiometer WIRA-C also for ozone measurements, and thus, we present a way to get additional scientific data out of our campaigns. We discuss the performance of our spectrometer configuration and its impact on retrievals of ozone. A cross-validation with other ozone instruments is not in the scope of this study. Instead, this study should motivate the inclusion of WIRA-C in future cross-validations.

## 2. Materials and Methods

### 2.1. Instrument and Spectrometer Configuration

The WIRA-C radiometer is a single-side-band heterodyne receiver, which uses a USRP (Universal Software Radio Peripheral) as a high resolution FFTS to measure the 142.17504 GHz rotational emission line of ozone with a high spectral resolution of 12 kHz over a bandwidth of 200 MHz. Figure 1 shows a schematic of the radiometer front-end and the back-end. The front-end consists of a 140 GHz Low Noise Amplifier (LNA), a single-side-band filter, and a sub-harmonic mixer that converts the Radio Frequency (RF) signal down to an Intermediate Frequency (IF) band centered at 3.65 GHz.

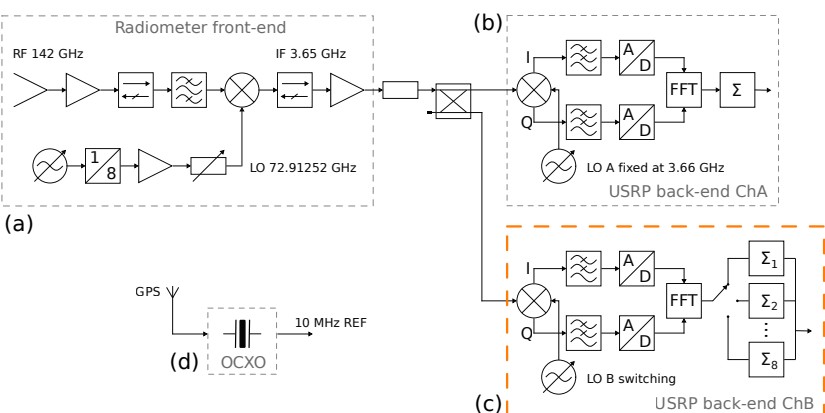

**Figure 1.** Schema of the WIRA-C radiometer front-end (**a**) and the back-end with spectrometer ChA (**b**) with fixed LO and changed ChB (**c**) with switching LO. All LO clocks are connected to the same GPS disciplined Oven Controlled Crystal Oscillator (OCXO) (**d**). IF, Intermediate Frequency.

The Universal Software Radio Peripheral (USRP X310 with two CBX-120 daughter-boards [7]) is used as the radiometer back-end; see Figure 1c,d. It has two independent channels (ChA and ChB), each of which embeds an In-phase and Quadrature (IQ) base-band converter and Analog-to-Digital Converters (ADCs) with a sampling rate of 200 MHz each. Due to the filter characteristics of the CBX-120 daughter-board input stage, only 160 MHz of the theoretically achievable 200 MHz are usable. We implement the FFT and accumulation on the internal Field-Programmable Gate Array (FPGA) using LabView-RT. After absolute-value-squared accumulation, the raw spectra are sent off to a general purpose computer for further processing. Both channels of the back-end are fed with the same IF signal, while ChA provides the high resolution spectrum for wind retrievals; we reconfigured ChB to

provide a wide spectrum with a total bandwidth of 1 GHz centered around the same region as ChA. An overview of the configuration is shown in Table 1.

**Table 1.** Overview of the channel configuration of the USRP used for this study. Both channels (ChA and ChB) are electrically equal, but ChB takes advantage of frequency scanning for larger Bandwidth (BW) at the cost of a lower total integration time.

|  | Technical BW | Usable BW | Resolution | Integration Time |
|---|---|---|---|---|
| ChA | 200 MHz | 160 MHz | 12.2 kHz | 10 s |
| ChB | $8 \times 200$ MHz | 1000 MHz | 61.0 kHz | 1.13 s |

While both channels have the same ADC with a sampling rate of 200 MHz, we use fast Local Oscillator (LO) switching to scan a broader band with ChB. This method switches the LO frequency of the base band converter in ChB fast enough to cover multiple adjacent spectral regions. We chose a total of 8 different LO frequencies and take the central 125 MHz of each spectrum. The combination of the spectra is done so that we have a small overlap of a few channels between the different parts. Then, we average over 5 channels to reduce noise, which resulted in 2048 channels per LO frequency. The details (binning, overlap, and bandwidth) are chosen so that the combined spectrum has 16,384 (=$2^{14}$, same as ChA) channels and a 1 GHz bandwidth, which gives a channel resolution of 61 kHz.

In theory, the integration time for ChB is exactly 8 times less than for ChA. However, in practice, the re-locking of the base-band converter LO takes some time, during which we do not acquire any ADC values. The ratio between actual integration time for ChA and ChB is 8.88 for 10 seconds of integration, meaning that we loose 1 s or 10% of the integration time. In absolute numbers, this means that an LO reconfiguration takes approximately 140 ms. This applies only to the data presented in this study, and we were since able to reduce this to less than 1 ms by optimizing the LO reconfiguration procedure.

## 2.2. Data Processing

We calibrate our spectrometer with a tipping curve scheme. This calibration method uses the observations under different elevation angles to calculate the zenith opacity and thus the brightness temperature of the sky in the zenith direction. One calibration cycle takes 2 min, which is well within the time span for which the receiver can be assumed to be stable, as presented in [4]. Together with the internal hot-load, this allows a hot-cold calibration. Notably, each FFT channel of the spectrum is considered separately. We use the zenith measurement and our slanted observation at a $40°$ elevation angle in the north and southward directions. The general scheme of this calibration method is described in [8], and its implementation for WIRA-C is given in [4].

Then, we perform a tropospheric correction of the calibrated spectra by estimating the tropospheric contribution using the left wing of the measured ozone emission line [4,9]. Using this estimation, we remove the tropospheric contribution from our spectrum. To account fully for the tropospheric contribution, the slope of the continuum of the water-vapor line at 182 GHz would also have to be considered. We do not correct for this in the tropospheric correction step, but leave it for the retrieval to deal with the linear offset in terms of a baseline.

To characterize the quality of our concatenated spectrum, we compare the measurements to the Atmospheric Radiative Transfer Model (ARTS) [10]. Firstly, this lets us subtract the contribution from the ozone line from our measurements, and we can compute the statistics of the residuals for the different parts of the concatenated spectrum. Secondly, we can give a first estimate of the performance of our setup with regard to ozone retrievals.

The retrieval includes the adjustment of the ozone profile and a baseline (constant and linear) to fit our measurement. We use the Optimal Estimation Method (OEM) from ARTS for the inversion. The OEM uses an a priori estimation with corresponding co-variances to regularize the inversion problem [11]. The ozone retrieval is performed for 81 pressure levels, logarithmically distributed between $10^2$ and $10^{-3}$ hPa. The ozone a priori data are the same as used in [4], namely the ozone

a priori data form the F2000 WACCM scenario from [12]. We impose a co-variance matrix with a diagonal of 1 ppm ozone VMR. The temperature data is taken from the ECMWF operational analysis, and we extend it with the F2000 WACCM data above 70 km altitude. The co-variance matrix of the spectrum is assumed to be diagonal (uncorrelated channels), and the variance is estimated by the Allan variance of the spectrum. We validate this estimation a posteriori against the variance of the residuals.

Once we have fitted the atmospheric model to our data, we compute the residuals and statistics for each block of the spectrum. We repeat the same evaluation method for different combinations of the spectra: (1) for the original spectrum from ChA, (2) for the concatenated spectrum from ChB, and (3) for a combination of data from ChA and ChB. For the combination of both channels, we take the spectrum from ChA and extend it with those parts from the broad-band ChB that are not covered by ChA itself.

In this study, we use data from the campaign in Bern that took place from March to May 2018. We selected this campaign because the data and methods used in this study might be of particular interest for the future validation of the GROMOS radiometer that is also located in Bern [13–15]. Specifically, we look at data for the night- and day-time of 7 May 2018 to 9 May 2018 and integrate over 12 h. For this measurement, the WIRA-C instrument was configured to look at a 40° elevation angle (same as GROMOS) instead of 22°, which is used for routine wind measurements. The WIRA-C instrument looks in four different cardinal directions (plus zenith) for wind measurements, but for this study, we only consider the northward measurement.

## 3. Results

Figure 2 shows a narrow spectrum (from ChA) and a wide spectrum (from ChB) colorized to give an impression of the concatenation process. The concatenation is done with a slight overlap, where one part blends into the next one. This is done for no specific reason, and we consider it to be an implementation detail with no bearing on the results. The spectra were acquired at nighttime 9 May 2018. The top panel of Figure 2 shows the ChA and ChB with a zoom on the line-center. Both channels show a matching brightness temperature. Note that from this figure, it is not clearly visible that ChA has less noise due to longer total integration time. We will later determine the noise from the residuals of a retrieval.

To quantify how well the spectra from the different channels match and also how well the spectra from the different parts of ChB match, we run a retrieval (and thus, fit an atmospheric model) on our data. As described above, we used three different spectra for our retrievals (ChA, ChB, and combined). The results from all these retrievals are shown in Figure 3, also for nighttime on 9 May 2018.

The increased bandwidth extends the sensitivity for ozone towards lower altitudes, whereas the high spectral resolution adds sensitivity in the mesosphere, as expected. For the combined retrieval, the measurement response is larger than 0.8 between approximately 21 km up to 70 km.

We also see an effect on the vertical resolution of the ozone profile, which is characterized by the Full Width at Half Maximum (FWHM) of the averaging kernels. For example, at an altitude of approximately 40 km, the averaging kernels from the retrievals from ChB have a FWHM of 14 km, which improves to 11 km when using the higher resolution of ChA. This difference is most notable at higher altitudes. Only below 30 km, the FWHM of the averaging kernels can profit from the wider bandwidth of ChB, which is able to capture more information from the pressure broadened emission line. This effect is caused by the better signal-to-noise ratio of the ChA measurement compared to that of ChB because of the longer integration time. The combined retrieval profits from the lower noise, as well as from the broader bandwidth at the lowest altitude levels. Another important measure of quality is the observation error. Here, the impact of the different spectral resolutions and noise are quite small, and the observational error is about 0.13 ppm for the whole altitude region covered.

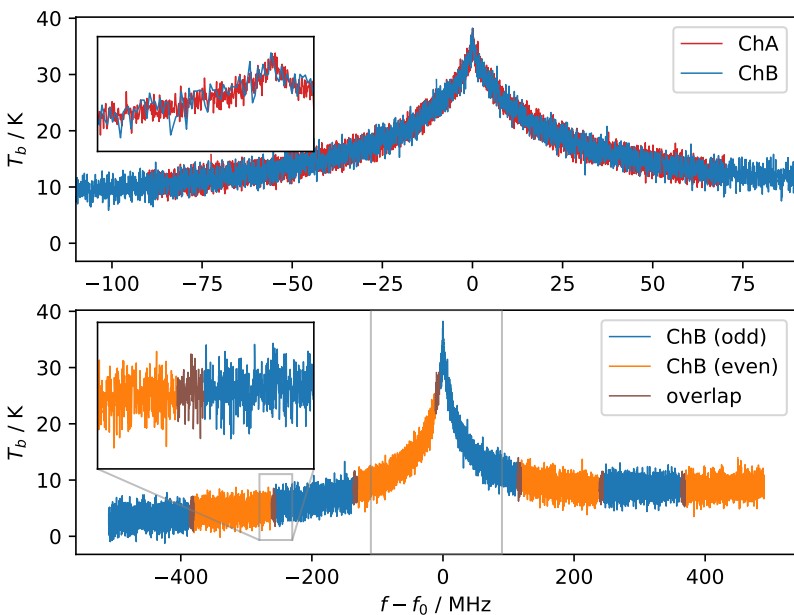

**Figure 2.** Spectra from the WIRA-C ChA (narrow) and WIRA-C ChB (wide) at an elevation angle of 40°. The bottom panel shows the different parts of the spectrum that were measured independently with alternating colors (odd/even). Frequencies are relative to the emission line center of ozone at $f_0 = 142.17504$ GHz.

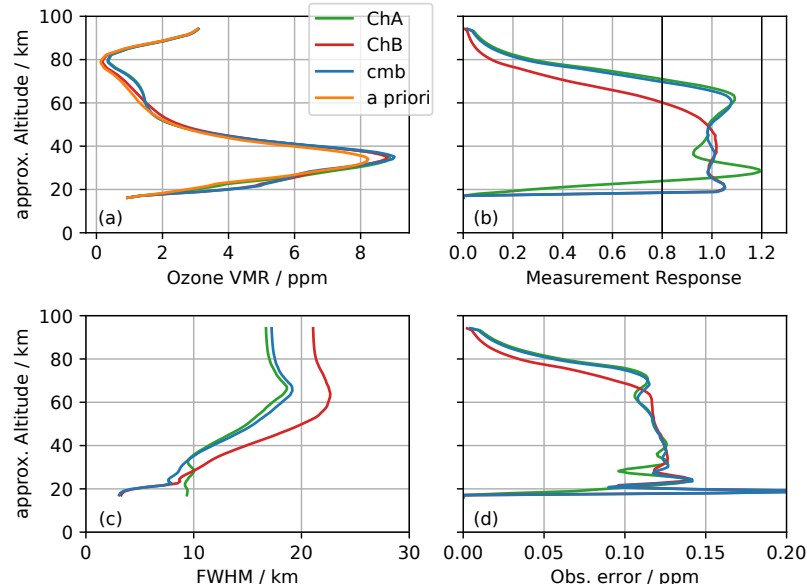

**Figure 3.** Ozone retrieval for three different spectra: ChA only, ChB only, and combined (cmb). Shown are the (**a**) ozone VMR, (**b**) the measurement response, (**c**) the full width at half maximum of the averaging kernels, and (**d**) the estimated observational error.

In general, the wider bandwidth only brings an improvement at or below approximately 30 km altitude. However, we would like to note that off-resonance measurements (that is, measurements at the line-wing) are important also for tropospheric correction and can be valuable for baseline correction.

Figure 4a shows the fitted spectrum together with the measurement and the baseline for the combined retrieval. There is a small slope visible in the baseline of approximately 3 K per GHz, which comes from the slope of the 182 GHz emission line of tropospheric water vapor that we did not correct for during tropospheric correction. Figure 4b shows the residuals (observed-minus-computed) of the

combined retrieval. There is no obvious baseline visible, and the transition from the broad spectrum to the high resolution one appears to be smooth.

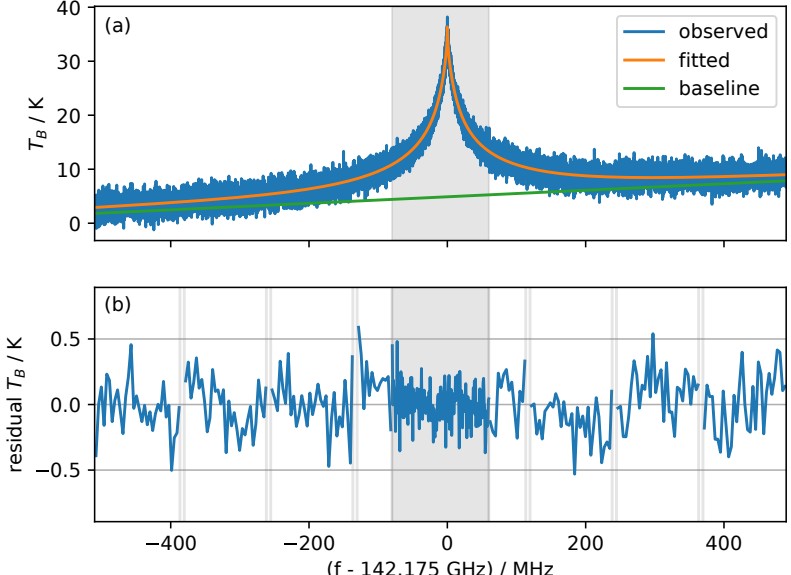

**Figure 4.** Measured and fitted spectrum for the combined (ChA and ChB) retrieval together with the baseline estimation (**a**). Residuals (**b**) are binned (64 FFTS channels), for each part separately. The spectral region from ChA is indicated by the gray area, and vertical gray lines indicate the boundaries of individual sub-spectra.

The residuals do not expose significant steps between the sub-spectra and are reasonably normally distributed as shown in Figure 5. Statistical evaluation shows no systematic dependency of residuals correlated with their origin (ChA or sub-spectra of ChB; no figure shown). The residuals of ChA have a standard deviation of 1.05 K, which is 1.31 times lower than the standard deviation of ChB residuals, which is 1.38 K. According to the radiometer formula, the relative noise between the two channels would be:

$$r_\sigma = \frac{\sigma_A}{\sigma_B} = \sqrt{\frac{\tau_A B_A}{\tau_B B_B}} = \sqrt{\frac{8.88 \times 1}{1 \times 5}} = 1.33 \tag{1}$$

which is very close to the observed value of 1.31.

Figure 6 shows the residuals of retrievals from ChB spectra for three days separated by daytime and nighttime (in total, six measurements). The distribution of the residuals is wider for days with a higher tropospheric opacity, because high humidity decreases the signal-to-noise ratio for the stratospheric signal. Some residuals expose a small step between the sub-spectra, for example on 8 May 2018 (day- and night-time) of about 0.2 K between Sub-spectra 7 and 8. Since the residuals can be assumed to be distributed with a standard deviation of $\sigma \approx 1.3$ K, we expect the difference between two channels to be smaller than $2\sigma \approx 2.6$ K in 66% of the cases. For the binned values shown in Figure 5, the expectation is that the difference between the two channels is smaller than 0.325 K in approximately 66% of the cases. For a future study on ozone retrievals with WIRA-C using the described spectrometer configuration, we thus recommend considering the possible steps between the sub-spectra of ChB in the quality control of the ozone retrieval. For example, one could consider measurements where the edges of the binned sub-spectra are more than 0.3 K apart for more than four of the seven boundaries to be invalid. The exact numbers would depend on the binning and distribution of residuals. In any case, the assumption of normally distributed residuals has to be checked first.

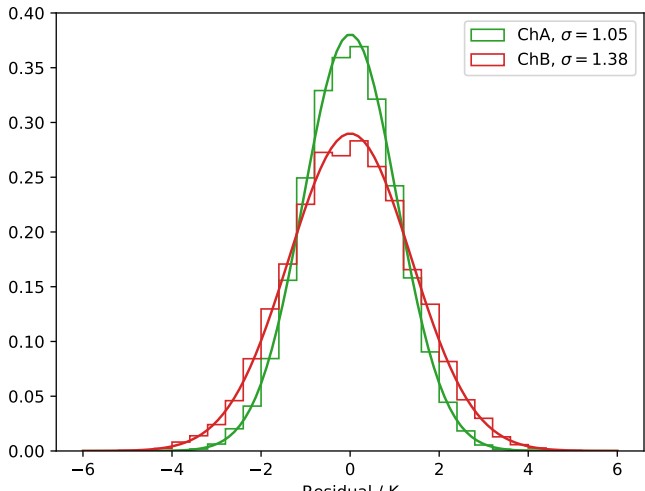

**Figure 5.** Histogram of the residuals for the different spectra (ChA only, ChB only, and combined) with the normal distribution corresponding to the data mean and standard deviation. Vertical gray lines indicate boundaries of individual sub-spectra. Measurements are annotated with tropospheric opacity $\tau$.

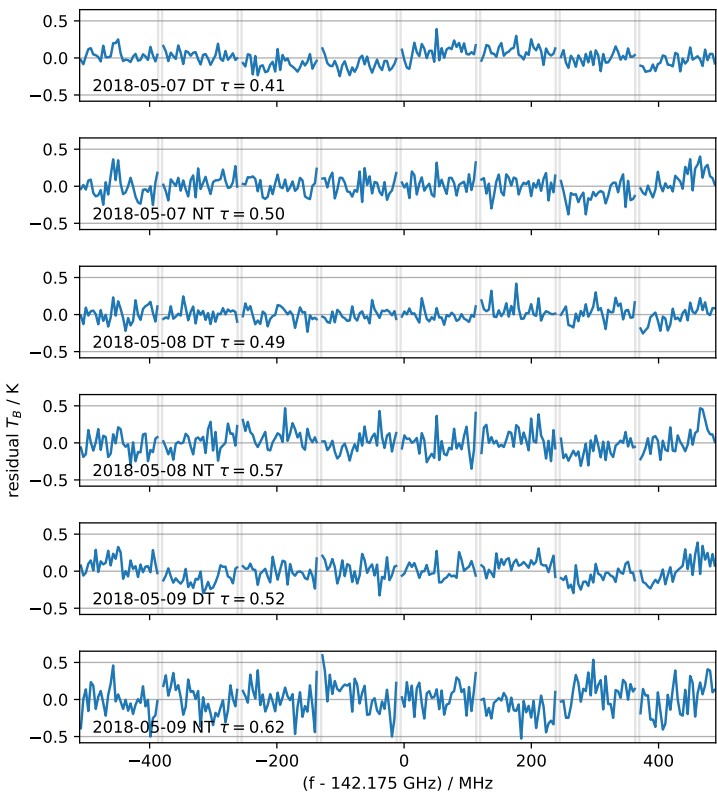

**Figure 6.** Residuals from six measurements taken during three days, daytime (DT) and nighttime (NT) obtained from the retrieval of ChB only. Residuals are binned (64 FFTS channels) for each part separately.

## 4. Outlook and Conclusions

We presented a method to measure broad-band spectra using off-the-shelf communications radio peripherals and demonstrated its use for radiometric observations of the atmosphere. While the method of fast LO switching itself is used in other applications, it is new to atmospheric remote sensing.

The LO switching reduced the integration time, and thus, the combination of a concatenated spectrum with a high resolution spectrum at the line-center was especially useful. At the line-wings, a larger bandwidth is preferred over high spectral resolution for ozone retrievals. This setup demonstrates how multi-purpose instruments can be built using off-the-shelf radio communications peripherals using SDR techniques.

We suggest a specific method to control the quality of the concatenated spectrum by rejecting measurements, where the binned values at the edges between the sub-spectra are off by more than $2\sigma$ at four or more of the seven edges (or equivalent).

An estimation of the retrieval quality showed that ozone retrievals could benefit from the broad spectrum, especially if the high resolution spectrum were combined with the broad-band spectrum at the line-center. In this specific case, we observed an improved vertical resolution of 11 km for the combined spectra over 14 km for the broad-band spectra, while keeping the measurement response larger than 0.8 down to 20 km (compared to 25 to 30 km for the high resolution spectrum from ChA only). While effective numbers are only valid for the presented case, the improvement is the result of the combination of the high resolution spectrum (used at the line-center) with the broad-band spectrum.

While the same hardware has already been used for operational wind measurements, this new method improves the capability to retrieve ozone profiles from the WIRA-C instrument. Since wind radiometers observe all four cardinal directions (north, south, east, west), they are also well suited to observe the spatial ozone distribution and gradients on a small scale. Together with wind observations, this facilitates gaining further insight into atmospheric dynamics and chemistry on small scales.

**Author Contributions:** Conceptualization, A.L. and A.M.; implementation/software A.L. and J.H.; investigation J.H. and A.M.; writing, original draft preparation, J.H.; writing, review and editing, J.H., A.M., and N.K.; supervision, N.K. and A.M. All authors read and agreed to the published version of the manuscript.

**Funding:** This research was supported by the Schweizerischer Nationalfonds zur Förderung der Wissenschaftlichen Forschung (Grant No. 200020-160048), the Staatssekretariat für Bildung, Forschung und Innovation (Grant No. 15.0262/REF-113./52107), and Horizon 2020 (Grant No. ARISE2 (653980)).

**Acknowledgments:** We acknowledge the effort of all technicians who contributed to this work or instruments used for this work, namely Nik Jaussi, Daniel Weber, and Adrian Jenk.

**Conflicts of Interest:** The authors declare no conflict of interest. The funders and the vendors of equipment and software had no role in the design of the study; in the collection, analyses, or interpretation of data; in the writing of the manuscript; nor in the decision to publish the results.

## Abbreviations

The following abbreviations are used in this manuscript (in alphabetical order):

| | |
|---|---|
| ADC | Analog-to-Digital Converter |
| ARTS | Atmospheric Radiative Transfer Simulator |
| BW | Bandwidth |
| ECMWF | European Centre for Medium-Range Weather Forecasts |
| FFT | Fast Fourier Transformation |
| FFTS | Fast Fourier Transform Spectrometer |
| FPGA | Field Programmable Gate Array |
| FWHM | Full Width at Half Maximum |
| GROMOS | Ground-based Remote Ozone Monitoring and Observation System |
| IQ | In-phase and Quadrature (converter) |
| LNA | Low Noise Amplifier |
| LO | Local Oscillator |
| OEM | Optimal Estimation Method |
| SDR | Software-Defined Radio |
| USRP | Universal Software Radio Peripheral |
| VMR | Volume Mixing Ratio |
| WACCM | Whole Atmosphere Community Climate Model |
| WIRA-C | WInd RAdiometer for Campaigns |

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
