# Peer review of "Frequency-Agile FFT Spectrometer for Microwave Remote Sensing Applications"

_atmosphere, doi:10.3390/atmos11050490_

Round 1
Reviewer 1 Report
In line 84 to 89 for the combined observation with ChA and ChB:
Is ChA integrated continuously while the the LO is switched or are these two measurments one with constant LO frequency and one with changing LO frequency?
If ChA is integrated with different LO frequencies, who are the merged spectra disentangled?
Figure 4: Look into more observational data for two things:
- Increasing Baseline in last subspectrum
- The residuals have in 5 of the 6 splicing regions some local extrema.
Figure 5: How do the Gaussians for the different subspectra compare to each other?
How are the differences between two spectra in the overlap region distributed?
How is the time stability (Allan variance) of the instrument?
Or is it not important for these kind of observations? Please explain.
Author Response
Dear Anonymous Reviewer,
please find our reply, among the replies to all Reviewers, in the attachment.
Best regards,
Jonas Hagen

Reviewer 2 Report
Dear Authors,
Please find my comments in the attachment.

Author Response

(The authors gave the same response as above.)

Reviewer 3 Report
This manuscript proposes and investigates the use of FFTS for atmospheric remote sensing. The WIRA-C Dopply wind radiometer is used as a test-bed and example, and compared to models. Overall this is a well-written and organized paper that provides a good pathfinder for a new avenue in atmospheric remote sensing. As is often the case in remote sensing manuscripts, particularly those dealing with new technology or applications, a comparison to modelling is a necessary substitute for actual validation, although comes with the usual drawbacks and caveats from a scientific standpoint.
The introduction and methods are clear and concise, while being appropriately thorough, and the authors raised all my immediate comments in the discussion. I would suggest that attention is paid to some mistakes and potential language corrections (detailed with line references below). I would also discourage the use of the abbreviations "approx." for approximately, and "fig." for figure throughout, unless journal style standard disagree with my preference.
Line 7: “approx.” to “approximately”
Line 21: “overview over” to “overview of”
Line 22: “available as” to “available in the form of”
Line 23: “which do not only” to “which not only”
Line 28: “approx.” to “approximately” (more examples of this throughout).
Line 29: “allows to retrieve” to “allows the retrieval of”
Line 30: “software defined radio” to “SDR?”
Line 32: “200 Mhz, that” to “200 Mhz, which” or “200 Mhz that”
Line 38: “chemistry the middle” to “chemistry of the middle” or “chemistry in the middle”
Line 45: “more scientific data” to “additional scientific data” if quantity is proposed to increase or “more robust data” if quality is proposed to increase.
Line 48: “single-sideband” to “single-side-band” (as in line 39), see also line 52.
Line 51: “Fig. 1” to “Figure 1”
Line 56: “Fig. 1” to “Figure 1” (more examples of this throughout)
Line 62: “are being sent off” to “are sent off”
Line 75: “some time during which” to “some time, during which”
Line 77: “meas” to “means”
Line 91: “campaign, because” to “campaign because”
Line 101: Perhaps rephrase to something like “which is an implementation detail with no bearing on the results” or similar?
Line 119: “larger” to “longer”
Line 120: “lowermost” to “lowest”?
Line 124: “note, that” to “note that”
Line 124: “that is measurements” to “that is, measurements”
Line 147: “this allows to gain” to “this allows us to gain” or “this facilitates” or similar
Line 147: “insight to” to “insight into” or “insight regarding”.
Abbreviations: There are other abbreviations used in the manuscript, such as SDR and FWHM.
Author Response

(The authors gave the same response as above.)

Round 2
Reviewer 2 Report
The authors have satisfactorily responded to all my questions and made the necessary changes to the manuscript. The current manuscript seems ready for publication as far as I can tell.